

# Freshwater-adapted sea bass *Dicentrarchus labrax* feeding frequency impact in a lettuce *Lactuca sativa* aquaponics system

Paraskevi Stathopoulou[1], Panagiotis Berillis[1], Nikolaos Vlahos[1,2], Eleni Nikouli[1], Konstantinos A. Kormas[1], Efi Levizou[3], Nikolaos Katsoulas[3] and Eleni Mente[1]

[1] Department of Ichthyology and Aquatic Environment, School of Agricultural Sciences, University of Thessaly, Volos, Greece
[2] Department of Animal Production, Fisheries and Aquaculture, School of Agricultural Sciences, University of Patras, Mesolonghi, Greece
[3] Department of Agriculture Crop Production and Rural Environment, School of Agricultural Sciences, University of Thessaly, Volos, Greece

## ABSTRACT

The aim of this study is to investigate the effect of three daily fish feeding frequencies, two, four and eight times per day (FF2, FF4, and FF8, respectively) on growth performance of sea bass (*Dicentrarchus labrax*)and lettuce plants (*Lactuca sativa*) reared in aquaponics. 171 juvenile sea bass with an average body weight of $6.80 \pm 0.095$ g were used, together with 24 lettuce plants with an average initial height of $11.78 \pm 0.074$ cm over a 45-day trial period. FF2 fish group showed a significantly lower final weight, weight gain and specific growth rate than the FF4 and FF8 groups. Voluntary feed intake was similar for all the three feeding frequencies treatmens ($p > 0.05$). No plant mortality was observed during the 45-day study period. All three aquaponic systems resulted in a similar leaf fresh weight and fresh and dry aerial biomass. The results of the present study showed that the FF4 or FF8 feeding frequency contributes to the more efficient utilization of nutrients for better growth of sea bass adapted to fresh water while successfully supporting plant growth to a marketable biomass.

## INTRODUCTION

The world's population growth, climate change, soil degradation, water pollution and food security management are some of the problems related to food production for human consumption. Aquaponic culture is an innovative and sustainable method for both fish and plant production and is environmentally friendly in relation to aquaculture fish and soil monocultures (*Tyson, Treadwell & Simonne, 2011*). The flexibility of an aquaponic system allows it to grow a large variety of vegetables, herbs, ornamental and aquatic plants to cater to a broad spectrum of consumers. Aquaponic products are organic and pesticide free, with a small environmental footprint (*Blidariu & Grozea, 2011*). Aquaponic growth contributes to water resource management, biodiversity conservation and energy savings (*Buzby & Lin, 2014*). In aquaponics, soil is not needed, and only a small

Corresponding author
Panagiotis Berillis, pveril@uth.gr

amount of water is required as the systems do not typically discharge or exchange water under normal operation but instead recirculate and reuse water very effectively. Thus, aquaponic systems can be set up in areas that have traditionally poor soil quality or contaminated water (*Blidariu & Grozea, 2011*).

Freshwater fish, especially tilapia species, carp, perch and catfish, are the main cultured species (*Bittsánszky et al., 2016*; *Nuwansi et al., 2016*; *Andriani et al., 2017*), along with some crustacean species such as *Cherax quadricarinatus* (*Diver & Rinehart, 2000*) and *Procambarus* spp. (*Saha, Monroe & Day, 2016*). Recent research by *Knaus & Palm (2017)* suggests multispecies cultivation is more efficient in an aquaponic system.

The ammonia oxidation to nitrite and nitrate via *Nitrosomonas* sp. and *Nitrobacter* sp. bacteria, is the basic principle of aquaponics. Ammonia released by fish through their metabolism is oxidized by nitration into nitrate ions (*Francis-Floyd et al., 2012*). Nitrate is useful for the plants and harmless (not toxic) for fish. Finally, the water is transferred back to the fish tanks and is nitrate-free (*Yavuzcan Yildiz et al., 2017*). Fish, plants and bacteria must coexist in balance in an aquaponic system. Therefore, the system type, the size of the filter, fish species, fish biomass and plant species and biomass should be carefully chosen. Proper combination of fish and plants leads to successful production without downgrading water quality (*Yavuzcan Yildiz et al., 2017*). The total biomass of fish should be calculated in comparison with plant biomass and the oxidizing capacity of the filter (*Endut et al., 2010*). If fish and plant biomasses are in appropriate proportions, the fish-produced daily ammonia is sufficient to meet 80% of the daily plant nutrition needs (*Rakocy et al., 2004*). Lower plant biomass will lead to nutrient accumulation in the systems, as a higher plant biomass will lead to slower plant growth (*Buzby et al., 2016*).

Fish feed supplies most of the essential nutrients required for optimal plant growth with the exception of Ca, K and Fe, which are usually inadequate and must be supplemented in aquaponic systems (*Rakocy, 2007*). Nitrogen and phosphorus in an aquaponic system are derived from fish food. Therefore, the rate of ammonia production depends on food quantity, its protein composition and the feeding frequency (*Cai, Wermerskirchen & Adelman, 1996*). The 5% (approximately) of feed is remained unconsumed by fish, and the rest 95% is ingested and digested. From the total amounts of nitrogen and phosphorus contained in the consumed food, only 30–40% is used by fish for their metabolism and growth (*Robaina et al., 2019*). The remaining 60–70% is released in the form of faeces, urine and ammonia (*Robaina et al., 2019*). The protein content of the fish diet differs between fish species and ages. A high protein content leads to better diet convertibility and improves fish growth (*Lazo, Davis & Arnold, 1998*). Approximately 1 kg of fish feed containing 30% crude protein releases approximately 27.6 g of N, and 1 kg of fish biomass releases 90.4 g of N and 10.5 g of P (*Robaina et al., 2019*). Many carnivorous fish species are less able to utilise dietary carbohydrates and cellulose contained in plant cells (*Jobling, 1995*). A plant-based protein fish diet can lead to higher plant biomass compared to an animal-based protein fish diet, but the growth rate of the carnivorous fish will be lower (*Medina et al., 2016*).

Nutritional status and feeding rhythm are factors that can affect the fish daily patterns of deamination of proteins and nitrogenous wastes. The feeding frequency and the feeding time affect ammonia production and the catabolism of proteins (*Kaushik, 1980*). According to *Gelineau, Medale & Boujard (1998)*, ammonia production and protein catabolism are lower in fish fed at dawn than in those fed at midnight. Among the different feed management practices proven to maximize the benefit of feeding, feeding frequency plays an important role in regulating the feed intake, minimize feed wastage and fish growth (*Cho et al., 2003*; *Silva, Gomes & Brandão, 2007*; *Biswas et al., 2010*). The optimal feeding frequency is very important to ensure optimum fish growth, survival, improved immunity and stress resistance (*Cho et al., 2003*). Feed loss and faecal waste is the largest contributor to solid waste in fish culture. The amount and relative composition of faecal material will be determined by the indigestible nutrients of the diet. An increased feeding frequency can lead to increased fish growth rates and increased amounts of excretion but lower food digestibility and water quality degradation (*Silva, Gomes & Brandão, 2007*; *Biswas et al., 2010*). Plants also show daily rhythms in nitrogen uptake. According to *Steingrover, Ratering & Siesling (1986)*, the nitrate concentration in the leaves increases during the night, as the uptake rate of nitrate by the roots increases at that time. Therefore, an increased feeding frequency contributes to more efficient plant nutrition (*Liang & Chien, 2013*).

Since the late 1980s, sea bass has become increasingly important in Europe, particularly for the Mediterranean region, with a steady increase in demand (*Federation of Greek Maricultures, 2019*). Sea bass (*Dicentrarchus labrax*) has not been used frequently in aquaponics. It is a euryhaline species, ideal for aquaponics with low salinity water in combination with edible or aromatic plants. Several studies have shown that sea bass can survive and grow in brackish water (*Pickett & Pawson, 1994*) and successfully be adapted to freshwater (*Nebel et al., 2005*). Increased mortality in freshwater adaptation can sometimes be detected, as fish in freshwater allocate more energy for osmoregulation than those in seawater. The success of sea bass to enter estuaries and river mouths may depend on a high degree on tissue resistance to changes in the plasma ion-osmotic status (*Allegrucci et al., 1995*; *Dendrinos & Thorpe, 1985*; *Jensen, Madsen & Kristiansen, 1998*). The habitat of fish plays an important role in their welfare and growth. The adaptation of a euryhaline fish from sea water to fresh water can affect its food digestibility (*Eroldoğan, 2004*). The frequency of feeding can affect the fish's nitrogen and energy utilization. In sea bass, a feeding frequency of 1–3 meals per day promotes better growth performance and food consumption rates (FCR) (*Güroy et al., 2013*), but this can vary with the time of year, fish size, fish feed and the production system.

The aim of this study was to evaluate the effect of 3 daily feeding frequencies (two, four and eight meals daily) on water quality, growth performance and histology of sea bass (*Dicentrarchus labrax*) adapted to freshwater in an aquaponic system. In addition, it examines which is the most efficient feeding frequency for sea bass in an aquaponic system that ensures the combined maximum growth performance of sea bass and lettuce plants (*Lactuca sativa*).
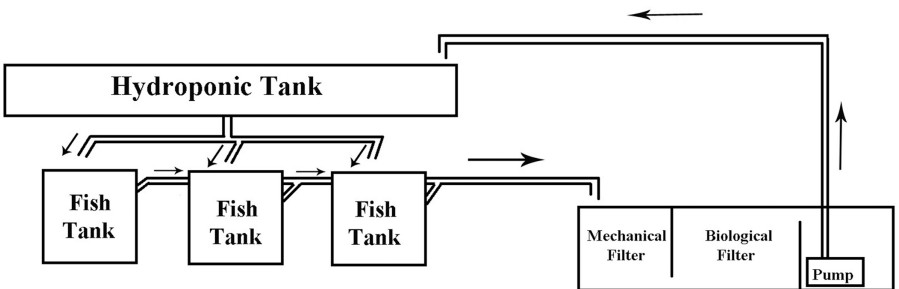

**Figure 1 Aquaponic system diagram.** Schematic diagram of the constructed aquaponics system. Arrows indicate the water flow.

# MATERIALS AND METHODS

## Aquaponic system and experimental set-up

Three (3) autonomous aquaponic systems with a total volume of 500 L each were constructed. Each system consisted of three glass fish tanks (50 cm × 50 cm × 50 cm) with a 100 L water volume each and a 26 L hydroponic cultivation tank (112 cm × 73 cm × 20 cm) paved with clay pebble (8–16 mm) substrate (Fig. 1, Fig. 2). Each aquaponic system' s nitrification process was enchased by a biological sump filter (100 cm × 50 cm × 48 cm) with a total volume of 184 L. Salinity was gradually decreased four units per week until it was stabilized to <1 ppt.

Sump filter construction was previously reported by *Vlahos et al. (2019)*. The sump filter was divided into three sections (Fig. 1). Suitable media, providing a specific surface area (SSA) for nitrifying bacteria colonization, covered the most of the filter area. The mechanical filter covered an area of 1,250 cm$^2$ and consisted of three layers of fibreglass material, creating in this way a 30 cm thick layer to retain the solid residues from the fish tanks (uneaten food and faeces). The biological filter covered an area of 2,150 cm$^2$ and was fixed by a mixed media of 20 L of porous cylindrical substrate K1 (10 mm diameter each), a 10 L ceramic ring (15 mm diameter each) and 10 L bioballs (30 mm diameter each). A pump (Aqua Medic OR 2,500 L/h, 38 W, 2.6 m $h_{max}$) was placed in the last part of the filter to supply the aquaponic system with water through the filter (Q = 6.27 L/min). Clay pebble substrate of the hydroponic tank also provided sufficient biofiltration, increasing the efficiency of the system. In each system, a high-pressure sodium 400 W lamp (Sylvania) was placed at a distance of 65 cm from the surface of the hydroponic tanks, ensuring this way the appropriate light for the plants. A winder photoperiod of 10 h light, 14 h dark was set up. An air-lift pump was used to recycle the water through a filter bed during the experiments (adjusted flow 1.53 L/min), thus creating a filtration speed (V) of 1.79 cm/min (*Vlahos et al., 2019*). The oxygen saturation levels were between 75% and 80%. Water flow from the hydroponic tank to the fish tank and to the sump filter was performed by gravity (Fig. 1). The setup period of the systems lasted 2 months to develop the biological filter. According to *Hirayama (1974)*, 40–60 days are necessary for the establishment of bacteria and the satisfactory oxidation of ammonia to nitrate ions.
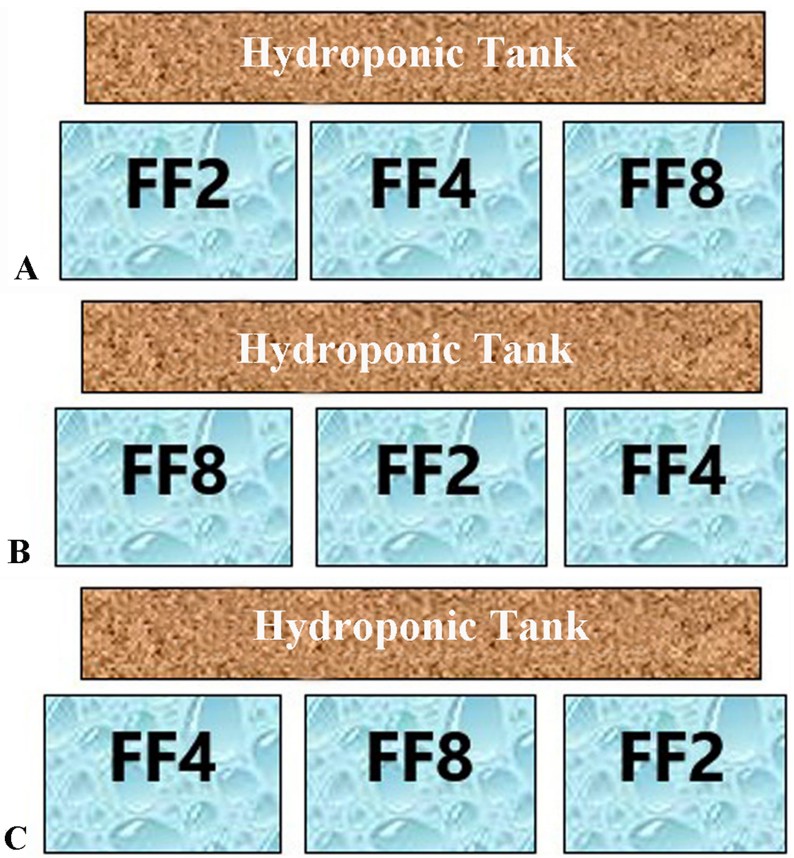

**Figure 2 Feeding frequencies.** Each of the three aquaponic system (A–C) was represented by all three feeding frequencies. The letters FF2, FF4, FF8 refer to feeding frequencies of two, four and eight meals/day respectively.

At the beginning of the experiment, an initial period of 24 h was used to permit any trace of chlorine to escape. Ten grains of a previously conditioned freshwater aquarium's filter bed were introduced to each aquaponic system, serving as inocula for nitrification bacteria. A total of 0.2 g of $NH_4Cl$ as an ammonia source was added and dissolved in each system. Water temperature (°C) and pH of fish tanks were recorded daily, while oxygen concentration (mg/L), electrical conductivity (mS/cm) and salinity were recorded every 3 days. Temperature, pH and the oxygen concentration were measured with multimeter sensors (HQ40d, Hach, Loveland, CO, USA); electrical conductivity was measured using a multimeter (CM35, Crison, Barcelona, Spain); and salinity was measured using an optical refractometer (ATC).

## Experimental design, fish rearing and plant growth conditions

Two hundred (200) juvenile seabass individuals of 1–2 g were transported from a local commercial fish hatchery, (SELONTA SA) located at Tapies-Pelasgia, at the Department of Ichthyology and Aquatic Environment (University of Thessaly), in special transport bags with oxygenation. Upon the arrival of the juvenile fish at the laboratory, they were placed for 3 h in aquariums filled with water of the same salinity (25‰) as the

transport water. Thereafter, salinity was gradually reduced by removing seawater and simultaneously adding fresh water to the desired salinity of 20 ‰. Every 7 days, the salinity was gradually reduced by 4 ppt. During adaptation, fish were fed to satiation twice a day. The temperature, pH and electrical conductivity (EC) were recorded daily. Total ammonia (mg/L), nitrites (mg/L) and nitrates (mg/L) in the water column were measured weekly (API AMMONIA $NH_3/NH_4^+$ TEST KIT, API NITRATE $NO_3^-$ TEST KIT and API NITRITE $NO_2^-$ TEST KIT were used respectively). Adaptation of fish to salinity <1 ppt lasted 60 days (*Marino et al., 1994*). Upon successful adaptation of the fish to fresh water, 19 fish were placed in each fish tank of the aquaponic systems and left for 15 days before the beginning of the experiment to permit their acclimation. Fish number was chosen according to *Hirayama (1974)* equation taking into account the equilibrium between the purification of the water and the pollution of the water as a result of the feeding and excretion of the fish. At the end of acclimation, their weight and total length were measured. Fish were placed in the aquaponic fish tanks in such a way that there were no statistically significant differences in initial weights and lengths among aquaponic fish tanks.

At the end of the acclimation period a total of 171 individuals juvenile sea bass, *Dicentrarchus labrax*, with an average body weight of 6.80 ± 0.10 g and an average body length of 8.62 ± 0.05 cm, were placed in the aquaponic fish tanks (9 fish tanks in total, 19 individual/tank). All experimental procedures were conducted according to the guidelines of the EU Directive 2010/63/EU regarding the protection of animals used for scientific purposes and were applied by FELASA accredited scientists (functions A–D). The experimental protocol was approved by the Ethics Committee of the Region of Thessaly, Veterinary Directorate, Department of animal protection-Medicines-Veterinary applications (n. 18402/05-09-2019). Experiment was conducted at the registered experimental facility (EL-43BIO/exp-01) of the Laboratory of Aquaculture, Department of Ichthyology and Aquatic Environment, University of Thessaly.

Fish were fed daily at 5% of their body weight with a commercial floating pellet diet (55% protein and 15% crude fat). Feed was distributed throughout the day (24 h) at three different feeding frequencies (FF) of two, four and eight meals/day over 45 days. The time period of 45 days was chosen for the best growth of the lettuce plants (*Andriani et al., 2017*). Each aquaponic system was represented by all three feeding frequencies (Fig. 1). Feeding was performed in a semi-automatic way. Feeding until 18:00 h (1 meal for FF2, 2 meals for FF4 and 4 meals for FF8) was performed by hand, and the other meals were performed with automatic feeders. The feeding rate was adjusted to fish weight every 15 days. Fish tanks were cleaned every day by siphoning, and uneaten food was removed. Daily food consumption per fish tank is calculated by the difference between the amount fed and the amount of uneaten feed collected (corrected for leaching losses) (*Vlahos et al., 2019*). At the end of the experiment, fish were anaesthetized with Tricaine methanesulfonate (MS 222), and their final fish body weights and lengths were measured.

Lettuce plants (*L. sativa* var. Musena) were grown in an unheated greenhouse until the 6-true-leaf stage. Five days prior to their transfer to the aquaponic system, Fe (Fe-DTPA), Ca (foliar application) and K (KOH) fertilization was performed. A total of 24 lettuce
seedlings were chosen, showing no statistically significant differences in their morphometric characteristics (height, number of leaves). Eight lettuce plants were evenly placed in each hydroponic bed, 20 cm apart. Plant number and density was chosen according to the dimensions of the hydroponic tank and the plant positions were carefully selected to ensure the homogeneity of the light environment; thus, each plant was exposed to 400–500 µmol m$^{-2}$ sec$^{-1}$ of photosynthetically active radiation (PAR). As the aquaponics systems were indoor systems, artificial light was used for the plants. The artificial light was supplied by a 400 W HPS lamp placed 65 cm above each growing area and accompanied by a timer for accurate control of the photoperiod (10 h light: 14 h dark). Plant height as well as the number of leaves were monitored every 15 days. No extra Ca, K or Fe was added to the aquaponic system.

## Water quality indicators

Ammonium ($NH_4^+$), ammonia ($NH_3$), nitrate and phosphate ions were monitored once a week before the daily first fish feeding. Water samples were taken at the water inlet point (GBin) and at the exit point (GBout) of the hydroponic cultivation tank.
All measurements were performed using a Hach DR3900 model photometer with special pre-weighted reagents.

The filter' s hydraulic loading ratio (HLR), recycled ratio (r), the hydraulic retention time of the water in the filter bed (HRT), the specific surface area of the filter (SSA) nad the volume of filter media ($V_{media}$) were calculated according to the equations described by *Endut et al. (2010)* and *Huguenin & Colt (2002)*.

HLR (m/day) = flow rate (Q)/total surface area of the trough

HRT (min)= (surface area water x depth x porosity of gravel trough/flow rate)

SSA (m$^2$/m$^3$) = Surface area of filter media/volume of the filter media

Vmedia (m$^3$) = surface area of the filter media/SSA

r=volume of recycled water/volume of the system

Production rate of ammonia nitrogen (PTAN) was calculated according to the below equation described by *Dediu, Cristea & Xiaoshuan (2012)*.

PTAN (mg/g fish/h) = ($C_e$ − $C_i$) * Q/W,

where: $C_e$, $C_i$ inlet and outlet ammonia concentration (mg/L), W: mean fish body weight in the tank (g), Q: flow rate(L/h).

## Fish and plants growth performance indicators

At the end of the 45-days, fish growth performance was calculated as below,

- Specific Growth Rate, SGR (%/day) = [(ln $W_{fin}$ − ln $W_i$)/$\Delta t$] × 100
- Weight Gain, WG (gr) = $W_{fin}$ − $W_i$
- Voluntary Feed Intake, VFI (% W day$^{-1}$) = 100 × food consumed (g)/[($W_{in}+W_{fin}$)/2 × $\Delta t$]
- Feed Conversion Ratio (FCR) = Feed consumed/WG

where $W_{in}$ and $W_{fin}$ are the initial and final weight of the fish respectively, and $\Delta t$ is the duration of the experiment in days.

Plant growth performance was calculated:

- Stem height (cm)
- Number of leaves
- Leaf fresh weight (gr) = Total fresh weight of leaves/Number of leaves
- Total fresh aerial biomass (gr) = Total fresh weight of leaves + Stem fresh weight
- Total dry aerial biomass (gr) = Total dry weight of leaves + Stem dry weight
- Total produced biomass (kg/m$^2$) = Total fresh weight of aerial part/cultivated area
- Root dry biomass (gr)

## Fish histology and gut microbiota structure

Euthanasia of animals followed the EU Directive 2010/63/EU and FELASA guidelines and performed through an overdose of Tricaine methanesulfonate (MS 222, 300+ mg/L). At the end of the experiment, five fish per tank were removed for histopathological examination, as previously described by *Vlahos et al. (2019)*. Fish were placed immediately on ice after euthanization. Samples of liver, midgut, kidney and gill were dissected from each fish. Tissue samples were fixed in Davidson' fixative for 24 h at 4 °C followed by dehydration in graded series of ethanol, immersion in xylol and embedding in paraffin wax. Thin sections of 4–7 μm were mounted, deparaffinized, rehydrated, stained with Hematoxylin-Eosin, mounted with Cristal/Mount and examined for alterations with a microscope (Axiostar plus Carl Zeiss Light Microscopy, Carl Zeiss Ltd, Gottingen, Germany) under a total magnification of 100× and 400×. A semi—quantitative grading system was used in order to quantify the histopathological alterations of the examined tissues (*Vlahos et al., 2019*). Severity grading used the following system: Grade 0 (not remarkable), Grade 1 (minimal), Grade 2 (mild), Grade 3 (moderate), Grade 4 (severe).

For midgut microbiota analysis, three fish at day 0 and three fish per feeding frequency treatment at days 0, 15 and 45 were sacrificed. Midgut samples were removed after dissection and DNA was extracted with DNA Mini kit (QIAGEN, Hilden, Germany). Bacterial diversity was assessed by amplification of the V3-V4 region of the bacterial 16S rDNA gene on the MiSeq Illumina platform 2 × 300 bp (MRDNA Ltd., Shallowater, city, TX, USA, sequencing facilities) using the primer pair S-D-Bact-0341-b-S-17 (5′-CCTACGGGNGGCWGCAG-3′) and S-D-Bact-0785-a-A-21 (5′-GACTACHVGGG TATCTAATCC-3′) (*Klindworth et al., 2013*). Polymerase Chain Reaction (PCR) was performed with HotStarTaq Plus Master Mix Kit (Qiagen, Valencia, CA, country) for 30 cycles of the following conditions: 94 °C (3 min), 94 °C (30 s), 53 °C (40 s), 72 °C (1 min) and a final extension at 72 °C (5 min).

Raw sequence reads were processed and analyzed using the MOTHUR software (v. 1.39.5) (*Schloss et al., 2009*) After trimming barcodes and primer sequences, quality control was performed through the 'screen.seqs' command and sequences were removed according to the following filtering parameters: length less than 250 bp, ambiguous bases, average quality score less than 25, and homopolymers longer than eight nucleotides. Thereafter, the remaining sequences were aligned against the SILVA 132 database (*Pruesse et al., 2007*). The VSEARCH algorithm was used to detect and remove chimeric

**Table 1 Water physicochemical parameters in the fish tanks over 45 days.**

|  | FF2 | FF4 | FF8 |
|---|---|---|---|
| pH | 6.75 ± 0.07[a] | 6.76 ± 0.70[a] | 6.77 ± 0.07[a] |
| EC (mS/cm) | 1.28 ± 0.01[a] | 1.28 ± 0.011[a] | 1.28 ± 0.11[a] |
| Salinity (ppt) | 0.64 ± 0.01[a] | 0.64 ± 0.01[a] | 0.64 ± 0.01[a] |
| $O_2$ (mg/L) | 8.59 ± 0.05[a] | 8.50 ± 0.06[a] | 8.52 ± 0.06[a] |

**Note:**
Data are expressed as mean ± S.E.M. ($n = 45$ for pH and $n = 15$ for EC, salinity, $O_2$). Means in a row followed by the same superscript are not significantly different ($p > 0.05$). EC: electrical conductivity.

**Table 2 Water quality of the three aquaponic systems over 45 days.**

| (mg/L) | System 1 | | System 2 | | System 3 | |
|---|---|---|---|---|---|---|
|  | GBin | GBout | GBin | GBout | GBin | GBout |
| $NH_4^+$ | 0.17 ± 0.03[a] | 0.09 ± 0.02[A] | 0.13 ± 0.02[a] | 0.07 ± 0.02[A] | 0.15 ± 0.04[a] | 0.08 ± 0.02[A] |
| $NH_3$ | 0.16 ± 0.02[a] | 0.08 ± 0.02[A] | 0.12 ± 0.02[a] | 0.06 ± 0.02[A] | 0.14 ± 0.04[a] | 0.07 ± 0.02[A] |
| $NO_3^-$ | 95.30 ± 16.88[a] | 81.44 ± 15.02[A] | 96.30 ± 23.51[a] | 86.16 ± 13.11[A] | 89.06 ± 14.93[a] | 72.03 ± 11.31[A] |
| $PO_4^{3-}$ | 37.270 ± 4.65[a] | 39.62 ± 3.61[A] | 40.40 ± 4.30[a] | 44.52 ± 3.48[A] | 37.44 ± 4.33[a] | 38.78 ± 3.98[A] |

**Note:**
Data are expressed as mean ± S.E.M. ($n = 7$). Means in a row per water inlet point (GBin) or exit point (GBout) of the hydroponic cultivation tank followed by the same superscript a or A are not significantly different ($p > 0.05$).

reads. Sequences were clustered into operational taxonomic units (OTUs) based on the average neighbor algorithm at a 97% sequence identity threshold (*Stackebrandt & Goebel, 1994*). High-quality OTU sequences were classified to different taxa according to the SILVA 132 database (*Pruesse et al., 2007*) with confidence value set above 80%.

## Statistical analysis

Values are presented as means ± standard error of the mean (S.E.M.). Data were tested for normality and homogeneity with Kolmogorov–Smirnov and Levene' s tests, respectively. To determine any significant differences between different feeding frequencies treatments, one-way ANOVA was used, followed by Tukey's post-hoc test. Independent t-tests were considered statistically significant at $p < 0.05$. Statistical analyses were carried out using the software package IBM SPSS Statistics V22.

# RESULTS

## Abiotic factors

Temperature was kept constant at 20 °C for each aquarium. The mean pH value was 6.75 ± 0.07, 6.76 ± 0.07 and 6.77 ± 0.70 for FF2, FF4 and FF8 respectively, while the dissolved oxygen levels were 8.59 ± 0.05 mg/L, 8.50 ± 0.06 mg/L and 8.52 ± 0.06 mg/L, respectively (Table 1). In all aquaponic systems the electrical conductivity was 1.28 ± 0.006 mS/cm while the average salinity was 0.64 ± 0.01 ppt (Table 1). There were no significant differences ($p > 0.05$) in the means of $NH_4^+$, $NH_3$, $NO_3^-$, $PO_4^{3-}$ and pH concerning the water quality in all of the 3 systems (Table 1, Table 2).

**Table 3 Functional characteristics of the filter in the aquaponic systems over 45 days.**

| Filter indexes | System 1 | System 2 | System 3 |
|---|---|---|---|
| HLR (cm/d) | $0.95 \pm 0.04^a$ | $0.96 \pm 0.05^a$ | $0.96 \pm 0.01^a$ |
| HRT (min) | $7.47 \pm 0.03^a$ | $7.46 \pm 0.01^a$ | $7.49 \pm 0.01^a$ |
| Q (m³/d) | $7.26 \pm 0.03^a$ | $7.34 \pm 0.03^a$ | $7.29 \pm 0.02^a$ |
| r | $0.90 \pm 0.01^a$ | $0.90 \pm 0.01^a$ | $0.90 \pm 0.01^a$ |
| $P_{TAN}$ (mg/L) | $6.69 \pm 0.66^a$ | $6.18 \pm 0.47^a$ | $6.71 \pm 0.75^a$ |
| SSA (m²/m³) | $362.9 \pm 36.19^a$ | $334.9 \pm 25.29^a$ | $363.82 \pm 40.56^a$ |
| $V_{filter\ media}$ (L) | $12.73 \pm 1.26^a$ | $11.75 \pm 0.88^a$ | $12.77 \pm 1.42^a$ |
| $V_{filter}$ (L) | 184.2 | 184.6 | 184.9 |

**Note:**
Data are expressed as mean ± S.E.M. ($n = 9$). Means in a row followed by the same superscript are not significantly different (ANOVA, $p > 0.05$). HLR, Hydraulic Loading Ratio; HRT, Hydraulic Retention Time; Q, Flow Rate; r, Volume of recycled water/volume of the system; $P_{TAN}$, Production rate of ammonia nitrogen; SSA, Surface area of filter media/volume of the filter media; $V_{filter\ media}$, Volume of filter media; $V_{filter}$, filter volume.

**Table 4 Growth performances of sea bass fed at three different feeding frequency over 45 days.**

| | FF2 | FF4 | FF8 |
|---|---|---|---|
| Survival (%) | $77.2^1 \pm 14.99$ | $96.50 \pm 1.75$ | $96.50 \pm 1.75$ |
| Initial weight ($W_i$, g) | $6.80 \pm 0.18^a$ | $6.78 \pm 0.17^a$ | $6.81 \pm 0.15^a$ |
| Final weight ($W_{fin}$, g) | $17.44 \pm 0.59^b$ | $20.29 \pm 0.77^a$ | $21.11 \pm 0.65^a$ |
| Weight gain (WG, g) | $10.66 \pm 0.40^b$ | $13.14 \pm 0.59^a$ | $13.85 \pm 0.50^a$ |
| Specific growth rate (SGR, %/day) | $2.11 \pm 0.05^b$ | $2.23 \pm 0.03^a$ | $2.36 \pm 0.02^a$ |
| Food Conversion ratio (FCR) | $1.6 \pm 0.51^a$ | $1.4 \pm 0.42^a$ | $1.3 \pm 0.37^a$ |
| Voluntary Feed Intake (VFI) | $2.72 \pm 0.06^a$ | $2.73 \pm 0.04^a$ | $2.64 \pm 0.01^a$ |
| Initial length ($L_i$, cm) | $8.54 \pm 0.09^a$ | $8.68 \pm 0.08^a$ | $8.64 \pm 0.08^a$ |
| Final length ($L_{fin}$, cm) | $11.66 \pm 0.13^b$ | $12.02 \pm 0.13^{ab}$ | $12.16 \pm 0.11^a$ |

**Note:**
[1] On the day 16th an unexplained fish mortality was observed (10 fish) for the FF2 group. This was probably due to anaesthesia fish handling. Consequently, it had no relation with the feeding procedures.
Data are expressed as means ± S.E.M. Means in a row followed by the same superscript are not significantly different ($p > 0.05$).

The $NH_4^+$, $NH_3$, $NO_3^-$ and $PO_4^{3-}$ fluctuation at the water inlet point (GBin) and at the exit point (GBout) of the hydroponic cultivation tank is shown in Fig. 2 respectively for all of the three systems.

Hydraulic loading rate (HLR), the recirculation rate (r), the retention time of the water into the filter bed (HRT), the flow rate (Q), ammonia production rate ($P_{TAN}$) the specific surface area of the filter bed (SSA), the volume of filter media ($V_{filter\ media}$) and the filter volume ($V_{filter}$) summarized in Table 3 were not statistically different ($p > 0.05$).

## Fish growth performance, histology and midgut microbiota

The fish growth performance is illustrated in Table 4. At the start of the study, there were no significant differences in the means of sea bass initial body weight (gr) and length (cm), (t-test, $p > 0.05$) for all the feeding frequencies groups (Table 4). At the end of the 45-days study period FF2 group showed significant lower final weight, weight gain, specific growth rate and final length than FF4 and FF8 groups, ($p < 0.05$), (Table 4). Voluntary feed

**Table 5 Severity score (0–4) for the observed histopathological alterations.**

| Feeding frequency per day | Liver | Midgut | Kidney | Gills |
|---|---|---|---|---|
| FF2 | 2 | 0 | 0 | 1 |
| FF4 | 2 | 0 | 0 | 1 |
| FF8 | 2 | 0 | 0 | 2 |

intake and FCR was similar for all the three feeding frequencies ($p > 0.05$), (Table 4). Survival rate for FF2, FF4 and FF8 was 77.2 ± 25.96%, 96.5 ± 1.75%, and 96.5 ± 1.75% respectively.

Liver histopathology of all the feeding frequency groups revealed mild (grade 2) accumulation of lipid droplets in liver cells with some of the nuclei of the liver cells to be pushed by the lipid droplets to the edge of the cells (Table 5, Fig. 3). Midgut and kidney microscopic examination showed no histopathological alterations (grade 0) in any of the feeding frequency groups (Table 4, Figs. 3, 4). Minimal (grade 1) gill histopathological alterations were detected in FF2 and FF4 groups (Table 5), while mild (grade 2) histopathological alterations were detected in FF8 group (Table 5). Epithelium detachment at the secondary lamellae and hyperplasia of primary lamellae were detected in some cases of all the 3 groups (Fig. 4).

A total of 2,506 bacterial operational taxonomic units (OTUs) were found in all samplings. The lowest number of OTUs occurred on day 0 (106 ± 36.0). The average number of OTUs on days 15 and 45 ranged between 232 ± 166.7 and 467 ± 129.0. Permutanional Analysis of Variance (PERMANOVA) of the OTUs relative abundance indicated that there were no statistically significant differences between sampling points and feeding frequency (Table 6).

## Plant growth performance

The plant growth characteristics are presented in Table 7. At the end of the 45-days study period, plants in all systems exhibited similar leaf fresh weight, total fresh weight of leaves, total fresh and dry aerial biomass (Table 7). Nevertheless, plants in system 2 showed inferior root growth and significantly lower number of leaves compared to system 1 and 3 (Table 7). Additionally, plants in system 3 significantly outweigh all the others in stem length.

# DISCUSSION

## Abiotic factors

In the present study an experimental aquaponic system for Mediterranean fish (sea bass) and a vegetable (lettuce) was studied for a duration of 45 days. To the authors' knowledge, this is one of a few studies using sea bass in aquaponic systems (*Waller et al., 2015*; *Nozzi et al., 2016a*; *Nozzi et al., 2016b*) and the first one to use three different feeding frequencies per day in the same aquaponic system. A successful aquaponic system provides important benefits, such as water quality control, high fish and plant growth performances,

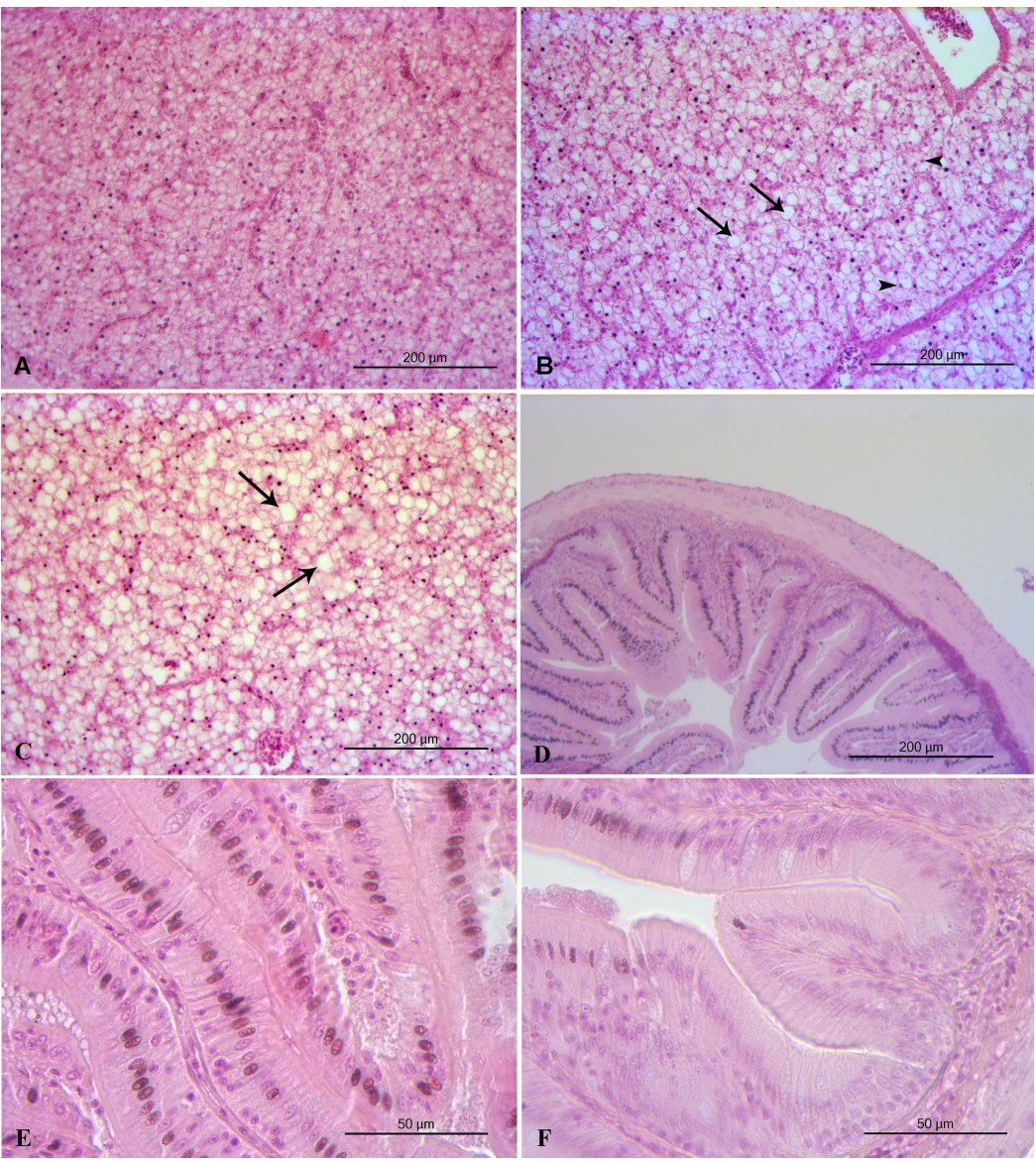

**Figure 3 Liver and midgut histopathology.** (A) FF2 group. Normal histological structure of liver. (B) FF4 group. Mild accumulation of lipid droplets in liver cells (arrows). Some of the nuclei of the liver cells are pushed by the lipid droplets to the edge of the cells (arrowheads). (C) FF8 group. Mild accumulation of lipid droplets in liver cells (arrows). Normal midgut structure. (D) FF4 group. Normal midgut villi structure, with normal enterocytes. (E) FF2 group. Normal midgut villi structure, with normal enterocytes. (F) FF8 group. Normal midgut villi structure, with normal enterocytes.

plant and fish disease management, and eliminating environmental impacts (*Vlahos et al., 2019*). Such systems require less than 5% of freshwater to be renewed due to evaporation or losses from daily functioning (*Hu et al., 2015*; *Nozzi et al., 2018*). Plant growth and production are indirectly related to feeding strategies, fish metabolic condition and microbial activity. Feeding rate and frequency affects nutrient availability in solution inside the system. Increased feeding frequency for fish contributes to more efficient plant

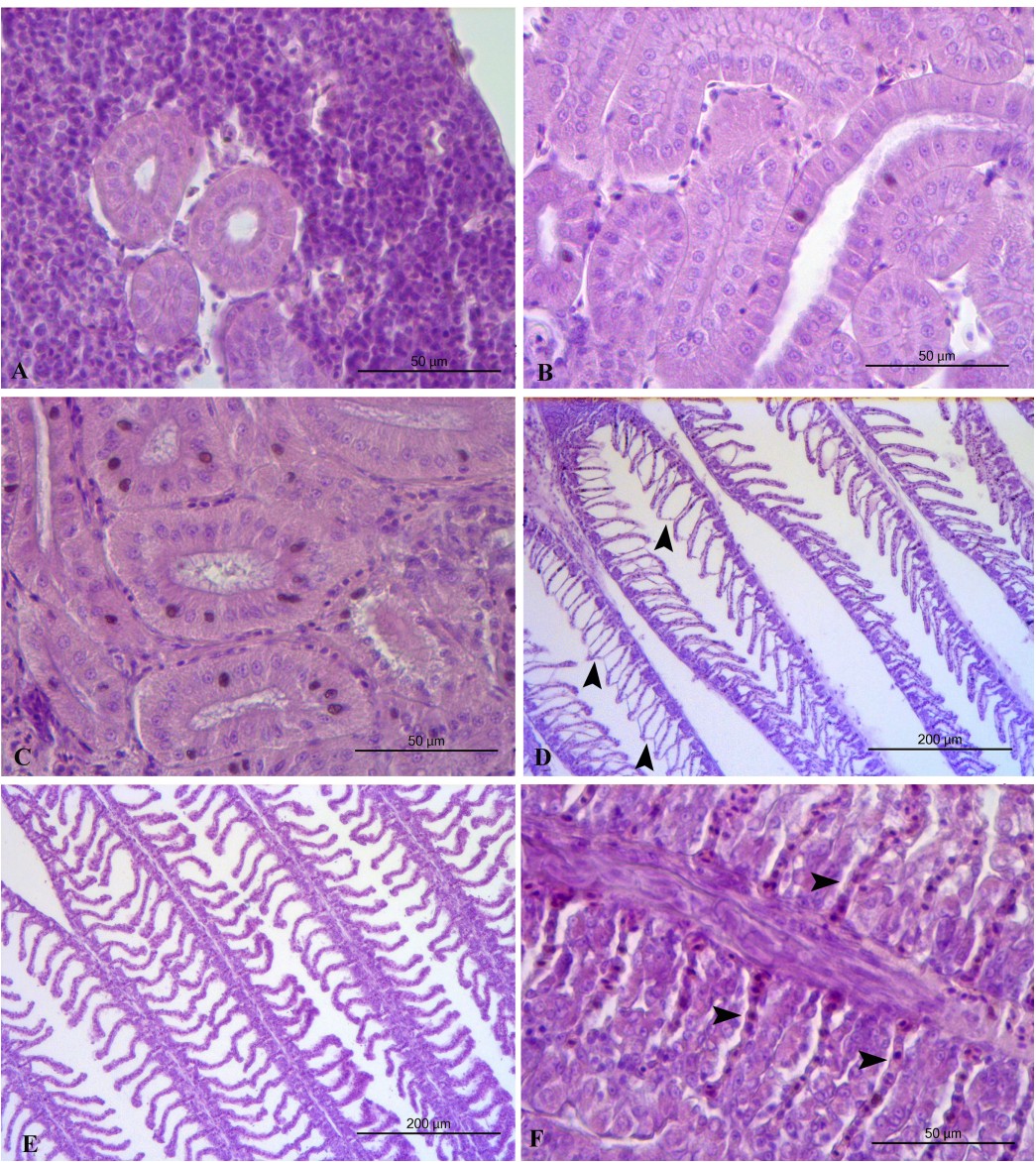

**Figure 4 Kidney and gills histopathology.** (A) FF2 group. Normal kidney structure. (B) FF4 group. Normal kidney structure. (C) FF4 group. Normal kidney structure. (D) FF2 group. Epithelium detachment at the secondary lamellae (arrowheads). (E) FF4 group. Normal gills structure. (F) FF8 group. Hyperplasia of primary lamella. Secondary lamellae (arrowheads) appeared trapped in the primary lamella.

nutrition (*Liang & Chien, 2013*; *Mohamed Abdelrahman, 2018*) as amounts of nitrate are available to the water for a longer period during the day.

The present study showed that the adaptation of sea bass in a fresh water aquaponic system together with cultivation of leafy vegetable lettuce is possible. Sea bass is an euryhaline specie. Direct transfer from sea to freshwater shows increased mortality (*Dendrinos & Thorpe, 1985*; *Cataudella et al., 1991*). However, fish gradually adapted over a period of one month (*Marino et al., 1994*) do not show any mortality.

**Table 6 Bray–Curtis similarity PERMANOVA results for between days (D) and times of feeding (td). F values upper half, *p* values lower half.**

|  | D0 | D15-FF2 | D45-FF2 | D15-FF4 | D45-FF4 | D15-FF8 | D45-FF8 |
|---|---|---|---|---|---|---|---|
| **D0** | – | 1.214 | 1.325 | 1.169 | 0.966 | 1.237 | 1.221 |
| **D15-FF2** | 0.297 | – | 1.590 | 1.051 | 1.162 | 1.579 | 1.330 |
| **D45-FF2** | 0.101 | 0.103 | – | 1.470 | 1.567 | 1.179 | 1.525 |
| **D15-FF4** | 0.101 | 0.295 | 0.099 | – | 1.299 | 1.550 | 1.155 |
| **D45-FF4** | 0.499 | 0.298 | 0.099 | 0.104 | – | 1.403 | 1.038 |
| **D15-FF8** | 0.104 | 0.105 | 0.298 | 0.104 | 0.097 | – | 1.141 |
| **D45-FF8** | 0.098 | 0.103 | 0.099 | 0.204 | 0.498 | 0.398 | – |

Note:
D0: Day 0; Dx-FFy: Sampling at the day x while feeding frequency was y times per day. i.e., D15-FF2: sampling at day 15 while feeding frequency was two times per day.

**Table 7 Plant growth performance.**

|  | System 1 | System 2 | System 3 |
|---|---|---|---|
| Survival (%) | 100 | 100 | 100 |
| Stem height (cm) | 7.73 ± 0.38[b] | 7.64 ± 0.35[b] | 9.63 ± 0.44[a] |
| Total fresh weight of leaves (gr) | 428.703 ± 47.58[a] | 375.23 ± 29.03[a] | 457.45 ± 21.39[a] |
| Number of leaves per plant | 43.88 ± 2.73[a] | 34 ± 0.87[b] | 43.63 ± 1.68[a] |
| Leaf fresh weight (gr) | 9.60 ± 0.66[a] | 10.98 ± 0.68[a] | 10.52 ± 0.41[a] |
| Total fresh aerial biomass (gr) | 463.04 ± 51.03[a] | 402.52 ± 31.66[a] | 496.06 ± 23.96[a] |
| Total dry aerial biomass (gr) | 23.20 ± 2.42[a] | 17.23 ± 1.50[a] | 21.47 ± 1.57[a] |
| Root dry biomass (gr) | 3.41 ± 0.54[ab] | 2.32 ± 031[b] | 4.64 ± 0.90[a] |
| Total produced biomass (kg/m$^2$) | 3.70 | 3.22 | 3.97 |

Note:
Data are expressed as means ± S.E.M ($n$ = 8). Means in a row followed by the same superscript are not significantly different ($p > 0.05$).

In an aquaponic system, the water temperature setting is dependent on the fish and plant species. Aquarium heater thermostats are used often in order to adjust the water temperature for the best growth of both plants and fish. For sea water-cultured sea bass, temperatures 19–22 °C show the maximum food utilization and growth rate (*Alliot, Pastoureaud & Thebault, 1983*; *Zanuy & Carrillo, 1985*). According to *Barnabé (1990)* and *Lanari, D'Agaro & Ballestrazzi (2002)*, higher weight gain can be achieved for sea bass at temperatures between 22–28 °C. In the present study, water temperature was kept constant at 20 °C, meeting the requirements of both sea bass and lettuce plants. The management of pH is also necessary in aquaponic systems. Plants, fish and bacteria require different pH ranges. Plants require a pH value between 5.5 and 6.5 to enhance the uptake of nutrients, and the optimal pH range for bacteria is 7.0–8.0, while the recommended pH for aquaculture is 6.5–8.5 (*Yavuzcan Yildiz et al., 2017*). So, an optimal pH range for an aquaponic system appears to be 6.5–7.0. pH > 7.0 can cause reduced solubility of phosphorus and micronutrients. Plant uptake of certain nutrients is restricted in the aquaponic environment (*Tyson et al., 2004*). In our study, pH showed a downward trend for all the three feeding frequencies with mean values of 6.75–6.77.

This downward trend is not unexpected, as the accumulation of nitrates (effective oxidation of ammonia) tends to make the aquatic environment more acidic. The mean value of pH is lower than 7.0 and within the tolerance levels for aquaponics. It is obvious that both pH and temperature are important parameters for the optimization of aquaponic production both for fish welfare/health issues and for plant needs.

Dissolved oxygen (DO) is the primary water quality consideration for aquaponic systems as in other aquaculture units. Oxygen levels of 7–8 mg/L ensure adequate ventilation for sea bass respiration (*Biswas et al., 2010*; *Güroy et al., 2013*), while oxygen levels >5 mg/L strengthen the plant's root system, nutrient uptake as well as the nitrification process (*Rakocy, Master & Losordo, 2006*; *Graber & Junge, 2009*). In general, the recommended limit for DO levels in fish culture is 6 ppm for cold water fish and 4 ppm for warm water fish (*Wedemeyer, 1996*).

The higher concentrations of $NH_3$, $NH_4^+$ and $NO_3^-$ at the inlet point (GBin) than at the exit point (GBout) of the hydroponic cultivation tank indicate that the lettuce plants absorbed nutrients through the water. According to *Von Wirén, Gazzarrini & Frommer (1997)*, the nitrogen form that plants absorb depends on the temperature. Low temperatures generally increased the reliance of plants on ammonium as a mineral nitrogen source. *Buzby & Lin (2014)* reported higher ammonia ($NH_4^+$ and $NH_3$) removal than nitrate by lettuce in an aquaponic system. *Xu, Tsai & Tsai (1992)* determined that ammonium was the preferred nitrogen source when nitrogen concentrations were low, while nitrate was preferred when concentrations were high. In the present study, all forms of nitrogen ($NH_3$, $NH_4^+$ and $NO_3^-$) at the exit point of the hydroponic cultivation tank exhibited lower concentrations after the 29[th] day (Fig. 2), indicating better absorbance from the lettuce plants after this day. In general, there were no major differences between the inlet point and exit point (Fig. 2). This is in accordance with the fact that lettuce has low ability to remove inorganic nitrogen (*Buzby & Lin, 2014*). Phosphate absorbance increased after the 32[nd] day (Fig. 2). According to *Buzby & Lin (2014)*, phosphate removal rates are increased when the plants are young and decrease over time. The type of substrate (clay pebbles) may affect the measurement of the nutrient concentrations at the exit point of the hydroponic cultivation tank. According to *Meinken (1996)*, nutrients can be absorbed (through diffusion) by clay pebbles and can be released back into the water circulation. This can be clearly seen from the 21[st]–28[th] days, when both nitrate and phosphate concentrations were higher at the exit point than the inlet point (Fig. S2). In the 0–14 day time period, the phosphate concentration was also higher at the exit point than the inlet point (Fig. 2). The gradual rise of nitrate levels proved the efficiency of the filter in oxidizing the produced ammonia. In the present study, the daily supply of 20–25 gr of fish food efficiently provides the necessary nutrients for plants. During the experiment, the water supply (Q) was adjusted to 6.27 L/min and the filtering speed (V) to 1.79 cm/min, ensuring the successful nitration and maximum efficiency of the filter (*Shete et al., 2016*).

The mean hydraulic loading rate (HLR) and hydraulic retention time (HRT) for all aquaponic systems ranged from 0.95 to 0.96 m/d and 7.46 to 7.49 min, respectively, indicating the efficiency of the filter performance and the nutrient removal efficiency.

The HRT impacts on the ammonia removal efficiency, alkalinity production, sulphate production and C/N ratio in the denitrification process (*Christianson et al., 2015*; *Suhr, Pedersen & Arvin, 2013*). HLR impacts on fish and plant production and nutrient removal (*Nozzi et al., 2016b*). According to *Chen, Ling & Blancheton (2006)* the best HLR for freshwater aquaponic systems is 1.28 m/day. *Endut et al. (2010)* reported that better growth performance of fish in a freshwater aquaponic system was observed at a higher HLR (2.56 m/d) than the HLR used in the present study. Nevertheless, *Vlahos et al. (2019)* reported that better growth performance of gilthead seabream and rock samphire was observed at an HLR of 1.84 m/d, which was higher than the HLR of the present study. The hydraulic loading rate (HLR) affects the production process of plants and fish and the daily nutrient removal efficiency and influences the contact time of the nutrients and microbial communities in the plants that grow in the bed (*Endut et al., 2010*; *Marino et al., 1994*). High values of HLR affect nutrient cycling in the hydroponic tank and reduce the nutrient contact time with bacteria in contrast with a lower HLR (*Gichana et al., 2019*).

According to *El-Sayed & Kawanna (2004)*, photoperiod is a factor that has a direct effect on the selected crop and does not exert a significant effect on fish growth. *Liang & Chien (2013)* came to a different conclusion. An increased photoperiod (24 h of lighting) leads to increased fish growth compared to 12 h of lighting (*Liang & Chien, 2013*). A high light intensity and long photoperiod can favour both plant and fish growth and can improve water quality (*Liang & Chien, 2015*; *Petrucio & Esteves, 2000*; *Rakocy et al., 1997*). In the present study, the photoperiod was adjusted to 10 h of light and 14 h of darkness, simulating the winter season and enhancing lettuce growth (*Lennard & Leonard, 2006*).

## Fish growth performance, histology and gut microbiota

During the 45-day trial period, the fish food was distributed throughout the day (24 h). Sea bass is an easily adjustable species in different feeding habitats (*Boujard et al., 1996*). According to *Sánchez-Vázquez et al. (1998)*, sea bass show seasonal preference in feeding hours, preferring the morning during spring and summer and the evening during winter. *Azzaydi et al. (2000)* showed higher SGR and lower FCR in night feeding during the winter months (0.26 ± 0.01%/day and 2.65 ± 0.08, respectively) in an RAS system compared to morning feeding (0.19 ± 0.01%/day and 3.73 ± 0.17, respectively).

The feeding frequency did not affect fish survival, with 77.2 ± 25.96%, 96.5 ± 1.75%, and 96.5 ± 1.75% survival being observed under the FF2, FF4 and FF8 treatments, respectively. On the day 16[th] an unexplained fish mortality was observed (10 fish) for the FF2 group. This was probably due to anaesthesia fish handling. According to *Gilderhus & Marking (1987)*, the margin between the effective and toxic concentrations of MS-222 tends to be narrow. Consequently, the observed mortality had no relation with the feeding procedures. In sea bass, a feeding frequency of 1–3 meals per day can deliver good growth and FCR performance (*Güroy et al., 2013*; *Kousoulaki et al., 2015*). For juvenile's sea bass (5.2–6.8 g) a feeding frequency of two times per day seems to be the minimum with good growth results and was followed to previous studies (*Gasco et al., 2016*; *Peres & Oliva-Teles, 1999*; *Peres & Oliva-Teles, 2002*). Nevertheless, in this study feeding frequency of 2,4 and 8 meals per day was tested in order to examine how it affects the daily

nitrate fluctuation for better plant nutrition in an aquaponic system. In a study by *Biswas et al. (2010)*, Asian sea bass fed one, two, three and four times a day in brackish water (3.2–4.1‰) showed the highest survival at three times (75.89 ± 4.17%) compared to other treatments. Similar results were reported for the fish species *Epinephelus tauvina*, *Aristichthys nobilis* and *Sparus aurata* (*Thia-Eng & Seng-Keh, 1978*; *Carlos, 1988*; *Goldan, Popper & Karplus, 1997*). *Vlahos et al. (2019)* working with *Sparus aurata* in an RAS aquaponic system under the salinities of 8‰ and 20‰ and a feeding frequency of three times per day, reported survival rates of 99% and 97%, respectively. In the present study, the survival rate of FF2, FF4 and FF8 feeding frequency treatment were higher than those for Asian sea bass and *Aristichthys nobilis* (*Biswas et al., 2010*; *Carlos, 1988*). The survival rates under FF4 and FF8 were similar to those reported for *Epinephelus tauvina* and *Sparus aurata* (*Marino et al., 1994*; *Thia-Eng & Seng-Keh, 1978*; *Goldan, Popper & Karplus, 1997*).

In an RAS and consequently in an aquaponic system, a properly selected diet must be managed in such a way as to meet the nutritional requirements of different fish and plant species. By selecting the appropriate food amount per day and appropriate feeding frequency, metabolic products (excretions) are reduced, fish growth is enhanced, and water quality ultimately improves (*Liang & Chien, 2013*). The removal of fish metabolic products (nutrients) from the water is directly related to the quantity of the provided diet, the feeding frequency and the food quality. Nitrogen content in fish faeces ranges (10% to 40%), depending on the nitrogen content of the provided diet and the fish type (*Van Rijn, 2013*). In the present study, fish were fed daily at 5% of their body weight with a commercial floating pellet diet (55% protein and 15% crude fat), showing good growth for all of the feeding frequency groups. These results suggest that the provided food amount was appropriate, and they are in agreement with those of *Eroldoğan (2004)*, where sea bass with an initial weight 2.6 ± 0.3 g cultured in seawater (40 ppm) and in fresh water (0.4 ppm) with six different feeding rates (2%, 2.5%, 3%, 3.5%, 4%, saturation) showed greater WG and SGR in fresh water and at a feeding rate of 3.5% until saturation. *Türkmen et al. (2012)* also showed that sea bass fed at 5% of their body weight four and eight times per day exhibited a higher SGR. In contrast, *Waller et al. (2015)*, working with sea bass fed daily to satiation, showed a lower SGR and FCR (1.5% and 0.93 respectively).

In aquaponic systems, increased feeding frequency seems to have positive effects on fish and plant growth. *Liang & Chien (2013)*, working in a tilapia-water spinach aquaponic system, reported that increasing feeding frequency increased both fish and plant production and lessened the accumulation of nitrogen and phosphorus nutrients in water. The same results were reported by *Mohamed Abdelrahman (2018)* while studying the effect of different daily fish feeding frequencies (one, two and three times per day) in a tilapia and lettuce aquaponic system. In the present study, the higher WG, SGR and were achieved at FF4 and FF8 (no significant differences were detected between these two feeding frequencies). FCR and voluntary feed intake did not differ among the three feeding frequencies ($p > 0.05$). Feeding four or eight times per day seems to have the best effects on fish growth. This result is in accordance with *Biswas et al. (2010)*, who showed that Asian sea bass (*Lates calcarifer*) cultured in brackish water had the best SGR when it was fed three or four times per day.

It is not clear if salinity is an important factor for the optimal growth of euryhaline species, as it is a disagreement among researchers if acclimatization to fresh water can cause a loss of appetite, increased mortality and decrement of conversion efficiency (*Allegrucci et al., 1995*; *Chervinski & Lahav, 1979*; *Silva & Perera, 1976*; *Peters, 1971*) or can cause similar or even better growth parameters than sea water (*Nozzi et al., 2016a*; *Eroldogan & Kumlu, 2002*; *Islam et al., 2020*; *Yilmaz et al., 2020*). Eroldogan and Kumlu (*Eroldogan & Kumlu, 2002*) showed that sea bass juveniles cultured in fresh water, 10 and 20 ppt grew better than those at 30 or 40 ppt. In a second experiment of the same study (*Eroldogan & Kumlu, 2002*), young sea bass grown in fresh water had higher WG than those grown in sea water, with a slightly higher FCR in sea water. *Vlahos et al. (2019)* did not detect differences in the growth performance of seabreams in two different salinities (8 ppt and 20 ppt). *Nozzi et al. (2016a)* showed higher WG and SGR for sea bass in fresh water than in sea water. Even at extreme temperatures, sea bass seems to grow better in low salinity water. According to *Islam et al. (2020)*, sea bass reared for 35 days followed by 10 days of extreme warm temperature (33 °C) showed higher weight gain and SGR at 12‰ and 6‰ salinity water than at 32‰. Weight gain and SGR were similar in 32‰ and 2‰ salinity (8.45 g and 9.42 g weight gain, respectively, and 2.03 and 1.93 SGR, respectively). In our study, SGR was 2.11, 2.23 and 2.36, while weight gain was 10.66 g, 13.14 g, and 13.85 g under FF2, FF4, and FF8, respectively. These values are higher than those reported by *Islam et al. (2020)*, probably because no temperature stress occurred. *Yilmaz et al. (2020)*, in a 60-day trial of the growth performance of sea bass in fresh water (0‰ salinity, 20 °C), reported a 1.1% SGR and 1.2 FCR, which SGR to be lower than the values in our study but FCR to be similar with our value in FF8 group.

Kidney is an important organ for the osmoregulation of euryhaline fish (*Greenwell, Sherrill & Clayton, 2003*; *Hammerschlag, 2006*). Fish like sea bass, trout, herring, and juvenile seabream show good adaptation to salinity changes, surviving this way in both seawater and freshwater. *Nebel et al. (2005)*, reported that sea bass juveniles lived in freshwater had smaller collecting ducts than those lived in seawater. *Vlahos et al. (2019)*, when adapting seabream to lower salinity (8 ppt), did not detect histopathological alterations of the midgut, smaller collecting ducts, granulomas or dilation of Bowman space in the kidney, hyperplasia of primary/secondary lamellae or epithelial detachment of the secondary lamella in gills, while liver histopathology showed inflammation and steatosis. In the present study, midgut and kidney microscopic examination showed no histopathological alterations, while liver showed mild accumulation of lipid droplets, and the gills showed mild epithelial detachment at the secondary lamellae and mild hyperplasia of the primary lamellae. Similar results for gills were reported in previous studies (*Laiz-Carrión et al., 2005*; *Masroor et al., 2018*), thus indicating the high plasticity and gill remodelling of sea bass adapted from seawater to freshwater. Lipid accumulation in the liver seems to be more extensive in sea bass living in sea water than in sea bass acclimatized to fresh water (*Nozzi et al., 2016a*).

The management of the aquaponics systems' water quality in order to meet the requirements of both reared fish and cultivated crops is not easy (*Yildiz et al., 2019*). According to *Yavuzcan Yildiz et al. (2017)*, a high level of suspended solids can affect the

health status of fish, provoking damage to the gill structure, such as epithelial detachment, hyperplasia, lamella fusion and reduction of epithelial volume. Feed waste includes both dissolved components (phosphorus and nitrogen-based nutrients) and suspended solids. Our results are similar to those reported by *Yavuzcan Yildiz et al. (2017)*. Uneaten food and faeces were removed daily by siphoning, but a breakdown in small particles still occurred. These particles are potentially dangerous and very difficult to collect. According to *Lekang (2013)*, the small particles will normally dominate in re-use systems.

Feeding frequency had no statistically significant impact ($p = 0.105$) on the structure of the midgut microbiota, indicating a minimal observable impact on the sea bass gut bacterial community. In this study, we analysed the resident midgut microbiota, i.e., bacteria that replicate inside the host's gut tissue cells, and not the transient bacteria associated with the animal's digesta (*Hammer, Sanders & Fierer, 2019*). Moreover, it has been recently shown that even nutritionally similar diets in sympatrically reared fish species cannot override host genetics in shaping the resident gut microbiota (*Nikouli et al., 2020*). Thus, the resident microbiota is expected to be less variable with pulses in the feed supply, as was the case in our study. However, small qualitative differences in the structure of gut bacterial communities have been reported for other fish after a short period after a single meal (*Mente et al., 2018*), and this remains to be investigated for freshwater-adapted sea bass.

## Plant growth performance

The successful cultivation of various plant species, including herbs, fruiting crops and leafy vegetables, in aquaponics has been well documented during the last decade of intensive relevant research. Lettuce is one of the most commonly studied species, mainly because it is a widely consumed vegetable worldwide with low to medium nutritional requirements, a short harvest period and its cultivation is convenient in terms of light and space (*Rakocy, Master & Losordo, 2006*). Many studies have examined the performance of lettuce in aquaponic systems and furthermore have compared it with hydroponics and complemented aquaponics (*Palm et al., 2019*). To the best of our knowledge, there are only two published experiments concerning the effect of the fish feeding frequency on lettuce growth (*Mohamed Abdelrahman, 2018*; *Rakocy et al., 1997*). In an early study, *Rakocy et al. (1997)* concluded that a higher fish feeding frequency had a positive effect on lettuce growth in aquaponics. In accordance with this finding, *Mohamed Abdelrahman (2018)* reported that an increased feeding frequency contributes to a more efficient nutrient supply to lettuce. The outcome was a 13.7% increase in lettuce production ($kg/m^2$) when tilapias were fed thrice per day in comparison with once per day.

Despite the large number of published works involving lettuce in aquaponic systems, it is usually difficult to attempt comparisons of growth responses, mainly due to different experimental set-ups and physicochemical parameters, which greatly affect the results. Under this framework, the results of the present study concerning total fresh biomass production (3.22–3.97 $kg/m^2$) are intermediate between the low lettuce production of 47.9 $g/m^2$ reported by *Castillo-Castellanos et al. (2016)* and the values of 4.97 $kg/m^2$ obtained in the study of *Lennard & Leonard (2006)*. In the first case, the experiment

included a tilapia-lettuce cultivation system (tilapia stocking density 7 kg/m$^3$, fed four times per day 3.5% of body weight), and in the second involved the cultivation of lettuce and Murray cod (feeding rate of 1–1.5% of body weight, 43% protein) for 21 days. The same growth period of 21 days in the work of *Dediu, Cristea & Xiaoshuan (2012)* yielded similar lettuce production as our system, though the latter lasted 45 days and was conducted with a six times-lower initial fish density. It is well established that different variants of the same plant species can react differently even though growth conditions are identical. *Andriani et al. (2017)* co-cultivated lettuce and mixed fish species (catfish and Nile tilapia fed with 31–33% protein, 4% of body weight) for 49 days. They reported final fresh aerial biomass similar to the results of the present study, yet the different lettuce variety resulted in discrepancies in other growth characteristics due to the different plant architecture.

## CONCLUSIONS

This study highlights the use of economic important euryhaline fish, such as sea bass, in a fresh water aquaponics system for the production of high-demand plants, such as lettuce. This combined cultivation can have a positive effect on the increase of food production and food security. Increased fish feeding frequency can have a positive benefit to the combined cultivation, as it can lead to improved fish growth parameters and to an increased plant biomass. The production of high commercial and nutritional value foods along with the achievement of high-quality food production through aquaponics systems may not be far away.

### Funding
This research has been co-financed by the European Union and Greek national funds through the Operational Program Competitiveness, Entrepreneurship and Innovation, under the call RESEARCH–CREATE–INNOVATE (project code: T1EDK-01153, Acronym: FoodOASIS). The funders had no role in study design, data collection and analysis, decision to publish, or preparation of the manuscript.

### Grant Disclosures
The following grant information was disclosed by the authors:
Greek national research Funds and the European Regional Development Fund: EPAnEK 2014-2020.

### Competing Interests
Konstantinos A. Kormas is an Academic Editor for PeerJ.

### Author Contributions
- Paraskevi Stathopoulou performed the experiments, analyzed the data, prepared figures and/or tables, and approved the final draft.

- Panagiotis Berillis conceived and designed the experiments, performed the experiments, analyzed the data, prepared figures and/or tables, authored or reviewed drafts of the paper, and approved the final draft.
- Nikolaos Vlahos performed the experiments, analyzed the data, prepared figures and/or tables, and approved the final draft.
- Eleni Nikouli analyzed the data, prepared figures and/or tables, and approved the final draft.
- Konstantinos A. Kormas analyzed the data, prepared figures and/or tables, and approved the final draft.
- Efi Levizou analyzed the data, prepared figures and/or tables, and approved the final draft.
- Nikolaos Katsoulas analyzed the data, authored or reviewed drafts of the paper, and approved the final draft.
- Eleni Mente conceived and designed the experiments, analyzed the data, prepared figures and/or tables, authored or reviewed drafts of the paper, and approved the final draft.

## Animal Ethics

The following information was supplied relating to ethical approvals (i.e., approving body and any reference numbers):

The experimental protocol was approved by the Ethics Committee of the Region of Thessaly, Veterinary Directorate, Department of animal protection-Medicines-Veterinary applications (n. 18402/05-09-2019).

## Data Availability

The raw measurements are available in the Supplemental Files.

## Supplemental Information

Supplemental information for this article can be found online at http://dx.doi.org/10.7717/peerj.11522#supplemental-information.

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
