# Peer review of "Freshwater-adapted sea bass Dicentrarchus labrax feeding frequency impact in a lettuce Lactuca sativa aquaponics system"

_PeerJ, doi:10.7717/peerj.11522_

## Round 0.1 · original submission · Minor Revisions

Reviewers have found your work has a lot of merit. Please, consider the reviewers' comments, and revise your work adequately. Look forward to your revised manuscript. Thank you.

·

Basic reporting

Manuscript is found scientifically sound. Author’s dedications are appreciable.
 Materials and methods are described in detail. But it’s difficult to read the whole
paragraph. Authors are requested to add a simple flow chart/diagram/table to present
the complete methods.
 Over all context writings are grammatically correct and easy to read.
 Article structure is followed by professional English.
 Figures and graphs are fine.
 Raw data are shared as much I checked.
 Results are relevant to hypothesis
 It’s a cordial request to rewrite the abstract and conclusion.

Experimental design

I observed that this manuscript contained original aims and objectives
 Research question and problem identification is clear.
 Methods are suggested to compile in a flow chart/diagram/table.
 Bacterial analysis need to explain detail.
 Manuscript is maintained with ethical standard as far I observed.
 Methods are described with enough details and data replications are present.
 Over all experimental design is accepted but need to explain some terms in details
that I mentioned in reviewed word file.

Validity of the findings

Literature is clearly stated.
 This research has scientific impact
 Results are statistically sound
 Conclusions are not well decorated. So, conclusion is needed to improve.
 Limitations of experiments are expected to describe.

Additional comments

This research work is appreciable and it’s has scientific value
 Please try to write and explain the methods simply with flow chart/diagram/tables.
 To explain bacterial analysis, you can also use flow chart.
 Flowchart/diagram are more comfortable and convenience to the readers.
 I am convinced with this research but need to improve methods and materials part. Probably you did
hurry in conclusion writing. I found your research is very interesting. So, I also expect a compact and
convincing conclusion.
 I found some texts with insufficient references and in some no references. Please follow the
reviewed article.
 Please mention the p value in text when you state your result statistically significant or not.
 Please revise all references.
 I tried my best to suggest for improving this article, please try to correct and develop as much you
can.

Reviewer 2 ·

Basic reporting

The reporting of the study is largely ok except for a few spelling edits which have been made in the attached review comments file. A largely good background/context and justification for the study have been provided. However, several gaps in literature references particularly in the materials and methods were identified and so highlighted in the attached file. Unfortunately, the raw data file did not open hence, I was unable to evaluate the data. The article structure is okay though I have some issues and consequent suggestions/recommendations for the figures and tables which are detailed in the attached file. In summary:
- keys/notes showing meanings of acronyms used in the figures and tables should be included
- Figures 2 and 3 are suggested to be considered as supplementary files since the contents are summarised in Tables 1 and 2.
- The Photomicrographs had missing plates/figures for some of the feeding frequency group representatives
- The quantitative histopathology assessment did not involve an 'n' value and hence not demonstrated statistical analysis of the derived dataset.
- Othe specific comments can be found in the attached file.
The results obtained were largely discussed including the limitations of the study which was good (unexplained dead fish in FF2).

Experimental design

This study aligns with the scope of this journal and demonstrates a rigorous research method and effort. However, I have the following requests for clarification:
In the introduction, the last paragraph, please include a statement on the potential use or benefit of the results based on the study aim described.
In the Materials and methods;
- Please state if there were replicates of the fish exposure tanks though it appears so considering an estimated 171 fish exposed at 19 fish per tank. Please state clearly the number of replicates (biological) per fish feeding group.
- Please state if the fish were randomly distributed into the experimental tanks. Also, please state the number of fish procured from the breeding farm initially which were adapted for 60 days. Also, were there any mortality experienced during the period of acclimation/adaptation of the fish before the commencement of the experiment?
- It is not clear how 24 lettuce plants at 8 plants per bed were distributed into the aquaponics system containing 171 fish at 19 fish per tank. Please see/clarify this in the methods and Figure 1.
Other comments and edits are contained in the attached pdf file.

Validity of the findings

The areas with needs for clarifications, further edits, and inclusions have been stated under experimental design. Unfortunately, I was unable to access the raw data, hence could ascertain this. The statistical analyses are largely ok except for needs for inclusions and clarifications in the tables which have been detailed above and contained in the attached file.

Additional comments

The study is largely well conducted and I find the results of importance and application in aquaponics which can be applied to other relevant fish species and plants. It demonstrates contributions to achieving food security, value for money, and reduction in water pollution.

Annotated reviews are not available for download in order to protect the identity of reviewers who chose to remain anonymous.

---

## Round 0.2 · accepted · Accept

Reviewers have considered your revised manuscript. Both have recommended acceptance. The editor agrees with their decision. A reviewer has provided annotated manuscript indicating few typographical edits for the authors to address. This can be addressed when the Proof PDF is being revised. The authors have benefited from the peer-review process. Thank you for finding PeerJ as your journal of choice, and look forward to your future scholarly contributions. Congratulations.

·

Basic reporting

-The title is now very clear and attractive.
- Professional English is used all over the manuscript.
- Enough literatures are cited and background of this study is quite strong now
- Related diagram is now inserted in materials and methods for better understanding so it's very easy to readers.
- Other figures and tables are well described.
- The article is professionally structured
- All questions are answered by authors satisfactorily.
- Relevant results are stated with significant value and hypotheses are clear now.

Experimental design

-Methods are described with proper figure, it's very appreciable.
- The aims and scope of this journal are original as far as I checked.
-Research questions are well answered with relevant references.
- Investigations are performed with enough technical efforts.

Validity of the findings

-Abstract and conclusion are now well written and much more convincing than previous.
- The result is statistically sound
-The discussion part is now more strong with enough references.

Additional comments

I highly appreciate your correction on manuscript. Now it is concrete and presentable to the readers. I am satisfied with your answers. The explanations and clarifications are well described in the manuscript which are required. I think now it's ready to publish. However, the final decision will be made by the editor.
Hope for the best.

Reviewer 2 ·

Basic reporting

The revised manuscript addressed the comments and edits suggested in the original review comments satisfactorily.

Experimental design

The comments raised in the original manuscript have been addressed by the authors. Particularly, references have been included for methods used, number of fish used and justifications have been provided, figures have been updated including tables. The methodology is described in sufficient detail to enable replication.

Validity of the findings

Clarifications, justifications and edits have been made following the original review comments and this is satisfactory.

Additional comments

Thank you for the edits and revisions done to your study. Please find attached the revised documents with just a few minor typo edits.

Annotated reviews are not available for download in order to protect the identity of reviewers who chose to remain anonymous.